# Identification of the SHREK Family of Proteins as Broad-Spectrum Host Antiviral Factors

**DOI:** 10.3390/v13050832

**Published:** 2021-05-04

**Authors:** Deemah Dabbagh, Sijia He, Brian Hetrick, Linda Chilin, Ali Andalibi, Yuntao Wu

**Affiliations:** National Center for Biodefense and Infectious Diseases, School of Systems Biology, George Mason University, Manassas, VA 20110, USA; dabbagh.deemah@gmail.com (D.D.); she3@gmu.edu (S.H.); bhetrick@gmu.edu (B.H.); lchilin@gmu.edu (L.C.); aandalib@gmu.edu (A.A.)

**Keywords:** SHREK, PSGL-1, CD34, PODXL1, PODXL2, CD164, MUC4, TMEM123, HIV-1, Influenza A, SARS-CoV-2, Ha-CoV-2

## Abstract

Mucins and mucin-like molecules are highly glycosylated, high-molecular-weight cell surface proteins that possess a semi-rigid and highly extended extracellular domain. P-selectin glycoprotein ligand-1 (PSGL-1), a mucin-like glycoprotein, has recently been found to restrict HIV-1 infectivity through virion incorporation that sterically hinders virus particle attachment to target cells. Here, we report the identification of a family of antiviral cellular proteins, named the Surface-Hinged, Rigidly-Extended Killer (SHREK) family of virion inactivators (PSGL-1, CD43, TIM-1, CD34, PODXL1, PODXL2, CD164, MUC1, MUC4, and TMEM123) that share similar structural characteristics with PSGL-1. We demonstrate that SHREK proteins block HIV-1 infectivity by inhibiting virus particle attachment to target cells. In addition, we demonstrate that SHREK proteins are broad-spectrum host antiviral factors that block the infection of diverse viruses such as influenza A. Furthermore, we demonstrate that a subset of SHREKs also blocks the infectivity of a hybrid alphavirus-SARS-CoV-2 (Ha-CoV-2) pseudovirus. These results suggest that SHREK proteins may be a part of host innate immunity against enveloped viruses.

## 1. Introduction

Mucins and mucin-like molecules are highly glycosylated, high-molecular-weight cell surface proteins that possess a semi-rigid and highly extended extracellular domain [1]. P-selectin glycoprotein ligand-1 (PSGL-1), a mucin-like glycoprotein [2,3,4] has recently been found to restrict HIV-1 infectivity [5] through steric hindrance [6,7]. Mechanistically, PSGL-1 was found to be incorporated into virion particles, which inhibits virion attachment to target cells [6,7]. The extracellular domain of PSGL-1 was found to be necessary for its antiviral activity [6]. PSGL-1 shares structural features with other mucins and mucin-like proteins. It has a heavily glycosylated, elongated extracellular domain that extends nearly 60 nm from the plasma membrane [8,9,10,11] and sterically hinders virus attachment to target cells when incorporated into virions [6,7].

To test whether molecules with a similar structure would also have the ability to block virus infectivity, we selected a group of cellular proteins, including CD43, TIM-1, CD34, PODXL1, PODXL2, CD164, MUC1, MUC4, and TMEM123. These proteins have diverse coding sequences, tissue expression patterns, and functionalities, but share a similar structural feature, namely a highly extended and heavily glycosylated extracellular domain. CD43 is a sialomucin transmembrane protein expressed at high levels on T lymphocytes, monocytes, and some B lymphocytes [12]. TIM-1 is preferentially expressed on Th2 T cells, and also contains an extracellular mucin domain and an N-terminal immunoglobulin (Ig)-like domain [13]. Both proteins have previously been shown to block HIV-1 infection of T cells [6,7,14]. CD34, the common surface marker of stem and progenitor cells [15], is also a member of the sialomucin family of proteins, which contain an extensively O-glycosylated extracellular mucin-like domain. The podocalyxin-like proteins 1 and 2 (PODXL1 and PODXL2) are CD34-related sialomucins and share a similar structure with CD34 [16], and are also markers of hematopoietic stem and progenitor cells [17,18]. CD164 is another sialomucin co-expressed with CD34 on the surface of human hematopoietic cells, and is structurally similar to CD34 [19]. We also selected two mucins, MUC1 and MUC4, which are transmembrane mucins with dense arrays of O-linked oligosaccharides attached to the threonine and serine residues in the extracellular domains [1]. TMEM123 (Porimin) is another mucin-like protein with high contents of glycosylated threonine and serine residues in its extracellular domain [20].

In this study, using HIV-1 infection as a model, we tested these mucins and mucin-like proteins for their ability to inactivate HIV-1 infectivity, and found that all these proteins can block the infectivity of HIV-1 through inhibiting virus particle attachment to target cells. In addition, we also demonstrated that SHREK proteins are broad-spectrum host antiviral factors that block the infection of diverse viruses such as influenza A. These results suggest that SHREK proteins may be a part of host innate immunity against enveloped viruses.

## 2. Materials and Methods

### 2.1. Cells and Cell Culture

HEK293T (ATCC), MDCK (ATCC), and HeLaJC.53 (kindly provided by Dr. David Kabat) cells were maintained in Dulbecco’s modified Eagle’s medium (DMEM) (Invitrogen, Carlsbad, CA, USA) containing 10% heat-inactivated FBS and 50 units/mL of penicillin and 50 µg/mL of streptomycin (Invitrogen). HEK293T(ACE2/TMPRSS2) cells (kindly provided by Virongy) were maintained in DMEM (Invitrogen) supplemented with 10% FBS and puromycin (1 μg/mL) and hygromycin B (200 μg/mL) as instructed by the manufacturer. HIV Rev-dependent GFP indicator Rev-A3R5-GFP cells (kindly provided by Virongy) were cultured in RPMI-1640 plus 10% FBS supplemented with 1 mg/mL geneticin (G418) (Sigma-Aldrich, Saint Louis, MO, USA), 1 μg/mL puromycin (Sigma-Aldrich) and 50 units/mL of penicillin and 50 µg/mL of streptomycin (Invitrogen).

### 2.2. Plasmid Transfection and Virus Production

The infectious HIV-1 molecular clone pHIV-1(NL4-3) was obtained from the NIH AIDS Reagent Program. pCMV3-PSGL-1, pCMV3-CD43, pCMV3-CD164, pCMV3-TMEM123, pCMV3-CD34, pCMV3-PODXL1, pCMV3-PODXL2, pCMV3-TIM1, pCMV3-MUC1, pCMV3-MUC4, pCMV3-Empty, pMUC1-HA, pTMEM123-HA and SARS-CoV-2 S, M, E, or N expression vectors were purchased from Sinobiological. The Ha-CoV-2(Luc) vector was described previously [21]. pCMV6-XL5-ICAM1, pCMV6-XL5-CD2, pCMV6-AC-ITGB2, pCMV6-XL5-CD62L were purchased from Origene. The pHW-NA-GFP(ΔAT6) reporter plasmid and the A/WSN/1933 H1N1-derived plasmids pHW2000-PB2, pHW2000-PB1, pHW2000-PA, pHW2000-HA, pHW2000-NP, pHW2000-NA, pHW2000-M, and pHW2000-NS were kindly provided by Dr. Feng Li. For HIV-1 virus production, HEK293T cells were cotransfected in a 6-well plate with 1 μg of pHIV-1(NL4-3) plus the indicated doses of pCMV3-CD164, pCMV3-TMEM123, pCMV3-CD34, pCMV3-PODXL1, pCMV3-PODXL2, pCMV3-TIM1, pCMV3-MUC1, pCMV3-MUC4 or pCMV3-Empty. Supernatants were collected at 48 h post cotransfection. For influenza A-GFP reporter particle assembly, HEK293T cells were cotransfected with pHW2000-PB2, pHW2000-PB1, pHW2000-PA, pHW2000-HA, pHW2000-NP, pHW2000-NA, pHW2000-M, and pHW2000-NS (0.25 μg each); pHW-NA-GFP(ΔAT6) (1.5 μg); and pCMV3-PSGL-1, pCMV3-CD43, pCMV3-CD164, pCMV3-TMEM123, pCMV3-CD34, pCMV3-PODXL1, pCMV3-PODXL2, pCMV3-TIM1, pCMV3-MUC1, pCMV3-MUC4, or pCMV3-Empty DNA (0.5 μg each) in a 6-well plate. Viral supernatants were harvested at 48 h. Hybrid alphavirus-SARS-CoV-2 (Ha-CoV-2) particles were produced by cotransfection of HEK293T cells in 6-well plates with SARS-CoV-2 S, M, N, and E expression vectors (0.3 µg each), Ha-CoV-2(Luc) (1.2 µg), and each individual SHREK-expressing vector or a control empty vector (1.6 µg). Ha-CoV-2 particles were harvested at 48 h post-transfection.

### 2.3. Virus Infectivity Assays

For the HIV-1 infectivity assay, p24-normalized HIV-1 particles produced in the presence of PSGL-1, CD43, CD164, TMEM123, CD34, PODXL1, PODXL2, TIM-1, MUC1, MUC4, or empty vector were used to infect Rev-A3R5-GFP cells (0.2 to 0.5 million cells per infection). The percentage of GFP+ cells was quantified by flow cytometry at 48 or 72 h post infection. The IC_50_ inhibition curves were generated using GraphPad Software. For the influenza A virion infectivity assay, influenza A-GFP reporter viruses were assembled in the presence of PSGL-1, CD43, CD164, TMEM123, CD34, PODXL1, PODXL2, TIM-1, MUC1, MUC4, or empty vectors. Virions were harvested at 48 h and used to infect MDCK cells (3 × 10^4^ cells per infection). GFP expression was quantified at 24 h post infection by flow cytometry. For Ha-CoV-2 infectivity assay, HEK293T(ACE2/TMPRSS2) cells were infected for 2 h with Ha-CoV-2 particles assembled in the presence of each individual SHREK-expressing vector or an empty vector. Infected cells were washed and cultured for 18 h. Cells were lysed with Luciferase Assay Lysis Buffer (Promega, Madison, WI, USA). Luminescence was measured on GloMax^®^ Discover Microplate Reader (Promega).

### 2.4. HIV Env Incorporation Assay

HIV-1 particles were produced by cotransfection of HEK293T cells with pHIV-1(NL4-3) DNA (1 μg) plus 100 ng of vector expressing CD43, CD164, TMEM123, CD34, PODXL1, or PODXL2, or 200 ng of vector expressing MUC1 or MUC4. Empty vector was used to keep the amount of DNA equal in each cotransfection. Virus particles were harvested at 48 h and purified through 10% sucrose gradient by ultracentrifugation (10,000× *g* for 4 h at 4 °C). Particles were resuspended in LDS lysis buffer (Invitrogen) and subjected to Western blot analysis.

### 2.5. Detection of SHREK Proteins in HIV-1 Particles

HIV-1 particles were assembled in 6-well plates by co-transfection of HEK293T cells with pHIV-1(NL4-3) DNA (1 μg) plus an empty vector or the vector expressing each individual SHREK protein (400 ng). For MUC1 and TMEM123, vectors expressing heamagglutinin-tagged MUC1 or TMEM123 were used. Particles were harvested at 48 h, normalized for HIV-1 p24, and subjected to immuno-magnetic capture as previously described [22]. Briefly, magnetic Dynabeads Pan Mouse IgG (Invitrogen) (1 × 10^8^ beads/0.25 mL) were conjugated with one of the following antibodies, mouse anti-PSGL-1 antibody (KPL-1) (BD Pharmingen), mouse anti-CD43 antibody (1G10) (BD Biosciences, San Jose, CA, USA), mouse anti-CD164 antibody (67D2) (Biolegend, San Diego, CA, USA), mouse anti-PODXL1 antibody (222328) (R & D Systems, Minneapolis, MN, USA), mouse anti-PODXL2 antibody (R & D Systems), mouse anti-CD34 antibody (563) (BD Biosciences) or mouse anti-HA tag (HA.C5) antibody (Abcam, Cambridge, UK) for 30 min at room temperature. After conjugation, antibody-conjugated beads were incubated with SHREK-bearing viral particles for 1 h at 37 °C. The complex was pulled down with a magnet, and washed with cold PBS for 5 times. Captured viral particles were eluted in 10% Triton x-100 PBS, diluted, and quantified by p24 ELISA.

### 2.6. Viral Attachment Assay

HIV-1 virus particles produced in the presence of PSGL-1, CD43, CD164, TMEM123, CD34, PODXL1, PODXL2, TIM-1, MUC1, MUC4, or empty vector were incubated with HelaJC.53 cells (prechilled at 4 °C for 1 h) at 4 °C for 2 h. The cells were then washed extensively (5 times) with cold PBS buffer and then lysed with LDS lysis buffer (Invitrogen) for analysis by Western blot. The attachment of HIV-1 virus produced in the presence of TMEM123 was also separately tested by pre-incubating the virus with the mouse monoclonal anti-TMEM123 antibody (297617) (ThermoFisher, Waltham, MA, USA) (25 μg/mL) at 37 °C for 1 h, followed by incubation of the virus with HeLaJC.53 cells at 4 °C for 2 h and lysis of the cells with LDS buffer.

### 2.7. Western Blots

Cells were lysed in LDS lysis buffer (Invitrogen). Proteins were denatured by boiling in sample buffer and subjected to SDS-PAGE, transferred to nitrocellulose membrane, and incubated overnight at 4 °C with one of the following primary antibodies: mouse anti-HIV-1 p24 monoclonal antibody (183-H12-5C) (NIH AIDS Reagent Program) (1:1000 dilution), human anti-HIV-1 gp41 antibody (2F5) (1:1000 dilution) (NIH AIDS Reagent Program), mouse anti-MUC1 antibody (HMPV) (BD Biosciences) (1:1000 dilution), mouse monoclonal anti-MUC4 antibody (1G8) (ThermoFisher) (1:1000 dilution), mouse monoclonal anti-TMEM123 antibody (297617) (ThermoFisher) (1:1000 dilution), or anti-GAPDH goat polyclonal antibody (Abcam) (1:1000 dilution). Membranes were incubated with HRP-labeled goat anti-mouse IgG (KPL) (1:2500 dilution) for 60 min at room temperature, or goat anti-human 800cw-labeled antibodies (Li-Cor Biosciences) (1:2500 dilution) for 30 min at room temperature. Chemiluminescence signal was detected by using West Femto chemiluminescence substrate (Thermo Fisher Scientific), and images were captured with a CCD camera (FluorChem 9900 Imaging Systems) (Alpha Innotech, San Leandro, CA, USA). Blots stained with infrared antibodies were scanned with the Odyssey infrared imager (Li-Cor Biosciences).

### 2.8. p24 ELISA

Detection of extracellular HIV-1 p24 was performed using an in-house p24 ELISA kit. Briefly, microtiter plates (Sigma-Aldrich) were coated with anti-HIV-1 p24 monoclonal antibody (183-H12-5C) (NIH AIDS Reagent Program). Samples were incubated for 2 h at 37 °C, followed by washing and incubating with biotinylated anti-HIV immune globulin (HIVIG) (NIH AIDS Reagent Program) for 1 h at 37 °C. Plates were then washed and incubated with avidin-peroxidase conjugate (R &D Systems) for 1 h at 37 °C, followed by washing and incubating with tetramethylbenzidine (TMB) substrate. Plates were kinetically read using an ELx808 automatic microplate reader (Bio-Tek Instruments, Winooski, VT, USA) at 630 nm.

### 2.9. Surface Staining

HEK293T cells were transfected with 400 ng of pCMV3-PSGL-1, pCMV3-CD43, pCMV3-CD164, pCMV3-CD34, pCMV3-PODXL1, pCMV3-PODXL2, pCMV3-TIM-1, or pCMV3-Empty DNA. At 48 h post transfection, 0.5–1 million cells were stained with one of the following primary antibodies: mouse anti-PSGL1 antibody (KPL-1) (BD Pharmingen), mouse anti-CD43 antibody (1G10) (BD Biosciences), mouse anti-CD164 antibody (67D2) (Biolegend), mouse anti-CD34 antibody (563) (BD Biosciences), mouse anti-TIM-1 antibody (219211) (R & D Systems), mouse monoclonal anti-PODXL1 antibody (222328) (R & D systems), or mouse monoclonal anti-PODXL2 antibody (211816) (R & D Systems), followed by staining with Alexa Fluor 488-conjugated goat anti-mouse secondary antibody (Invitrogen).

### 2.10. SARS-CoV-2 S-Antigen ELISA

The SARS-CoV2-S antigen kit was purchased from Sinobiological (KIT40591). The ELISA procedure was performed according to the manufacturer’s manual.

## 3. Results

### 3.1. Inactivation of HIV-1 Virion Infectivity by Mucins and Mucin-Like Proteins

Previous studies have demonstrated that deletion of the highly glycosylated extracellular domain of PSGL-1 abolished its antiviral activity [6]. The extracellular domain of PSGL-1 consists of 14–16 decameric repeats (DR) with multiple O-glycosylated threonines (30%) and prolines (10%) [23,24]. DR plays a pivotal role in elongating and strengthening the extracellular N-terminal domain [24]. We found that when the DR domain was deleted from the N-terminus, the anti-HIV activity of PSGL-1 was also abolished (Figure 1A,C) [6]. These results suggest that the heavily glycosylated, large extracellular domain of PSGL-1 plays a critical role in blocking virion infectivity [6,7].

To determine whether mucins and mucin-like proteins with a similar extracellular structure would also inhibit HIV-1 virion infectivity, we selected and tested a group of mucins and mucin-like proteins, including CD43, TIM-1, CD34, PODXL1, PODXL2, CD164, MUC1, MUC4, and TMEM123 (Figure 1A). We cotransfected them individually with HIV-1(NL4-3) DNA into HEK293T cells to assemble viral particles, and then quantified the infectivity of the virions produced by infecting an HIV-1 Rev-dependent indicator cell, Rev-A3R5-GFP [6,25,26] (using an equal level of p24). The presence of each of these proteins in virus producer cells blocked the infectivity of the HIV-1 virions (Figure 1D).

To determine whether transmembrane proteins with large extracellular domains would similarly inhibit HIV-1 infectivity, we selected several non-mucin proteins with various extracellular repeat domains. Both CD2 and ICAM-1 are members of the immunoglobulin (Ig) superfamily of proteins that contain heavily glycosylated extracellular Ig domains (2 and 5 Ig domain repeats for CD2 and ICAM-1, respectively) [27]. ICAM-1 has been shown to enhance virion infectivity in an ICAM-1/LFA-1-dependent fashion [28,29,30,31]. L-selectin and E-selectin are members of the selectin family of cell adhesion molecules that share a similar extracellular structure with a variable number of consensus repeats (2 and 6 for L- and E-, respectively) [32]. We also selected integrin beta chain-2 (ITGB2), which has a large extracellular domain with integrin epidermal growth factor like repeat domains (I-EGF repeats 1 to 4) [33]. Each of these proteins was similarly cotransfected with HIV-1(NL4-3) DNA into HEK293T cells to assemble viral particles. Virions released were harvested and quantified for infectivity in Rev-A3R5-GFP cells (using an equal level of p24). As shown in Figure 1E, none of these proteins, except for E-selectin (Appendix A), was found to block the infectivity of HIV-1 virions. This is in contrast with the mucin and mucin-like proteins that we selected; all of these mucin and mucin-like proteins were found to block HIV-1 infectivity (Figure 1D). These results demonstrate that the ability of mucin and mucin-like proteins to block HIV-1 infectivity is not a shared property of transmembrane proteins expressed in virus producer cells.

We further performed experiments to quantify and compare the anti-HIV activity of the mucin and mucin-like proteins (Appendix A). We observed dosage-dependent inhibition of HIV-1 by all of them (Figure 2A,B, Appendix A), and the 50% inhibition dosages were determined (Figure 2C). Among these proteins, CD34, the common stem cell maker, had the strongest inhibition, blocking HIV-1 virion infectivity at an IC_50_ of 2.33 ng (Figure 2C). The MUC1 vector was less effective, with an IC_50_ of 216.06 ng (Figure 2C). Based on the antiviral activities and shared structural feature of these proteins, we conveniently named them the Surface-Hinged, Rigidly-Extended Killer (SHREK) family of virion inactivators.

### 3.2. Effects of SHREK Protein Expression on HIV-1 Virion Release

To elucidate the anti-HIV mechanisms of SHREK proteins, we first examined the effects of SHREK expression on virion release in virus producer cells. Previous studies of PSGL-1 have demonstrated that expression of PSGL-1 in virus producer cells had minimal effects on HIV-1 virion release [6]. To examine effects of SHREK proteins on HIV-1 virion release, we cotransfected HEK293T cells with HIV(NL4-3) proviral DNA (1 μg) plus each of the SHREK protein expression vector at varying inputs from 0.5 to 400 ng (Figure 3). The release of virions in the presence of each SHREK protein was quantified. All of the proteins except TIM-1 had minor effects on virion release at doses below 100 ng (Figure 3). CD34, PODXL1, and CD164 [34] inhibited HIV-1 virion release only at high dosages (100 ng and above). TIM-1 potently inhibited virion release at doses as low as 5–10 ng (Figure 3 and Appendix A), consistent with a previous report [14]. While not inhibiting virion release at low dosages, all of these proteins, except for MUC1 (IC_50_, 216.05), effectively blocked virion infectivity with an IC_50_ at approximately 10 ng and below (Figure 2B,C). These results suggested that, for most of these SHREK proteins, inactivating virion infectivity rather than blocking virion release is a major mechanism.

### 3.3. Virion Incorporation of SHREK Proteins and Effects on HIV-1 Env Incorporation

The anti-HIV activity of PSGL-1 is mainly attributed to its incorporation into virion particles that sterically hinders virion attachment to target cells [6,7]. In addition, virion incorporation of PSGL-1 also inhibits the incorporation of the HIV-1 Env protein [6,35]. To study the anti-viral mechanisms of SHREK, we examined virion incorporation of SHERK proteins. For this purpose, we adopted a previously established particle pull-down assay [22], using magnetic beads-conjugated anti-SHREK antibodies to pull down virion particles that express SHREK on the surface (Figure 4A). Magnetically purified virions were quantified for the presence of HIV-1 p24 in the virions. We were able to demonstrate that the anti-PSGL-1 antibody selectively pulled down p24+ virion particles produced from the PSGL-1 expression cells, but not from the cells co-transfected with the empty vector, recapitulating previous findings that PSGL-1 is incorporated into HIV-1 virion particles [5,6,7]. Using this pull-down assay, we were able to demonstrate that similar to PSGL-1, the anti-CD43, -CD164, -PODXL1, -PODXL2, and -CD34 antibodies selectively pulled down virion particles produced from these SHREK expression cells (Figure 4A,B), demonstrating virion incorporation of SHREK proteins. The anti-TMEM123 antibody had a weak ability to pull down virion particles produced from the TMEM123 expression cells. We were not able to use the commercial anti-MUC1 and anti-MUC4 antibodies to pull down virion particles. To overcome these technique difficulties, we replaced TMEM123 and MUC1 with hemagglutinin-tagged TMEM123 or MUC1 to assemble viral particles, and then used magnetic beads-conjugated anti-hemagglutinin antibody to pull down virion particles. Using this approach, we were able to demonstrate virion incorporation of TMEM123 and MUC1 (Figure 4A,B).

We further examined effects of SHREK incorporation on HIV-1 Env incorporation. Virion particles were assembled in the presence or absence of SHREK proteins, and then analyzed by western blots for HIV-1 Env gp41 and p24. As shown in Figure 4C, all SHREKs, except for TMEM123, had low-level inhibition of HIV-1 Env incorporation to various degrees; while TMEM123 did not affect Env incorporation, PODXL2 had the strongest inhibition.

### 3.4. SHREK Proteins Inhibit Virus Particle Attachment to Target Cells

We and others previously reported that PSGL-1-mediated inhibition of virion infectivity occurs mainly through virion incorporation, leading to PSGL-1-mediated steric hindrance of particle attachment to target cells [6,7]. We performed a similar virion attachment assay using particles produced from cells expressing each individual SHREK protein. As shown in Figure 5A, as a control, particles produced from the PSGL-1-expressing cells were highly impaired in their ability to attach to target cells. Similar strong impairment was observed for virions produced from the CD43-, CD34-, and PODXL2-expressing cells. Of note, these four proteins (PSGL-1, CD43, CD34, and PODXL2) have the strongest inhibition of HIV-1 among SHREKs, with an IC_50_ around 2 ng (Figure 2C). The other proteins (CD164, PODXL1, MUC1, and MUC4) also inhibited virion attachment to varying degrees. Interestingly, TMEM123 did not significantly inhibit HIV-1 virion attachment to target cells, although it inhibited viral infection (Figure 2B,C). One possible reason is that TMEM123 itself may interact with surface proteins, thereby promoting non-productive attachment of virus particles to target cells. Indeed, when the binding assay was performed in the presence of an anti-TMEM123 antibody to block possible TMEM123 interaction with cell surface proteins, virion attachment was decreased (Figure 5B).

### 3.5. SHREK Proteins Are Broad-Spectrum Host Antiviral Factors

PSGL-1 has been shown to be a broad-spectrum host antiviral factors [6]. To further determine whether these SHREK proteins also possess broad-spectrum antiviral activities, we tested their ability to block other viruses, including influenza A and a new hybrid alphavirus-SARS-CoV-2 pseudovirus [21]. To assemble influenza A virus, eight vectors expressing each of the segments of the influenza A/WSN/33 (H1N1) genome plus a GFP-reporter vector were cotransfected with individual SHREK proteins into HEK293T (Figure 6A). Viral particles were harvested and used to infect target MDCK cells. As shown in Figure 6B, the presence of each of the SHREK proteins in virus-producer cells inhibited the infection of target cells by the virions released. However, the degree of inhibition was different. MUC1, PODXL1, and MUC4 had the strongest inhibition of influenza A virus, although MUC1 and MUC4 were less effective against HIV-1. CD164 and TMEM123 had the weakest inhibition of influenza A. These results demonstrate that SHREK proteins are broad-spectrum and can block the infectivity of multiple viruses. Nevertheless, it is apparent that for each individual SHREK, its antiviral potency can vary among different viruses, with the differences likely related to possible viral antagonisms, the localization of SHREKs and sites of viral budding, and other unidentified factors.

To test possible effects of SHREK on SARS-CoV-2 infection, we assembled a newly developed hybrid alphavirus-SARS-CoV-2 particles (Ha-CoV-2) [21] in the presence of individual SHREKs, using PSGL-1 as a control. Ha-CoV-2 is a non-replicating SARS-CoV-2 virus-like particle, composed of SARS-CoV-2 structural proteins (S, M, N, and E) and an RNA genome derived from an alphavirus vector. Our recent study has shown that PSGL-1 can block SARS-CoV-2 S protein virion incorporation, virus attachment, and Ha-CoV-2 infection of target cells [36]. We performed similar experiments on the inhibition of Ha-CoV-2 infection by SHREKs, and found that in addition to PSGL-1, CD164, TIM-1, MUC1, and MUC4 also inhibited the infection of Ha-CoV-2 virus (Figure 6C); The MUC4 vector had the strongest inhibition among these SHREKs. However, we did not find inhibition of Ha-CoV-2 by CD43, CD34, PODXL1, PODXL2, and TMEM123 (Appendix A), although they are potent inhibitors of HIV-1 (Figure 2B,C).

We further examined possible effects of SHREK proteins on Ha-CoV-2 S protein incorporation and virion release by quantifying the amounts of S in the supernatant (Appendix A). Expression of MUC4 greatly reduced the S protein concentration, likely through blocking virion release or S incorporation; expression of MUC1 or PSGL-1 also led to a roughly 50% reduction of S in the supernatant [36]. For the rest of SHREK proteins, their expression had minor effects on virion release or S protein incorporation.

## 4. Discussion

In this report, we identified a group of proteins, namely SHREK, that share similar structural characteristics and the ability to broadly inactivate virion infectivity (Figure 1). We also quantified their antiviral activity in a range of doses (from 0.5 ng to 400 ng of SHREK expression vector), using HIV-1 infection as a model, and observed a strictly dosage-dependent inhibition of HIV-1 virion infectivity (Figure 2). The physiological relevance of SHREK proteins such as PSGL-1 in restricting HIV-1 virion infectivity has been previously confirmed in HIV-1 infection of primary blood CD4 T cells, and by shRNA knockdown of PSGL-1 in human CD4 T cells [6]. The slight reduction of endogenous levels of PSGL-1 in transformed and primary CD4 T cells led to an enhancement of virion infectivity, demonstrating that the presence of endogenous PSGL-1 can affect virion infectivity [6].

Although we mainly focused on proteins with mucin and mucin-like domains, we expect that SHREK proteins will likely include other proteins, such as E-selectin (Appendix A), with diverse molecular structures. The antiviral activities of SHREK can be achieved through at least three different mechanisms: (1) blocking virion release, e.g., TIM-1 inhibition of HIV-1 release [14]; (2) inhibition of virion incorporation of viral attachment proteins, e.g., PSGL-1 inhibition of HIV-1 gp160 incorporation [6]; (3) virion incorporation of SHREK that blocks progeny virus attachment to target cells through steric hindrance (e.g., PSGL-1 inhibition of HIV-1 virus attachment to target cells [6,7]). Our previous studies have also shown that viruses such as HIV-1 can evolve antagonisms to counteract the blockage imposed by SHREK proteins. For example, HIV-1 uses the accessory proteins Vpu and Nef to degrade and downregulate PSGL-1 on CD4 T cells [5,6]. However, it remains to be determined what other antagonisms may exist, by which different viruses antagonize individual SHREKs.

Previous studies have shown that several of the SHREK proteins, such as PSGL-1 [5], CD43 [37], CD164 [34,38], TMEM123 [39], MUC1 [40], and MUC4 [41], are induced by interferons. However, SHREK proteins in general are likely different from interferon-induced restriction factors, and can be constitutively expressed in specific cell populations. Interestingly, we noticed that multiple SHREKs (CD34, PODXL1, PODXL2, and CD164) identified in this study are heavily expressed on the surface of stem and progenitor cells. In particular, CD34, the common surface marker of stem and progenitor cells, is one of the most potent SHREKs against retroviruses such as HIV-1 in our study (Figure 2C). It is likely that these SHREKs are a part of host innate antiviral mechanisms that limit retroviral replication in critical cell populations such as stem and progenitor cells. It has been known that HIV-1 can enter and express genes in CD34+ multipotent hematopoietic progenitor cells (HPCs), but viral replication is limited [42]. However, viral replication can occur in HPCs with GM-CSF and TNF-α treatment, which induces myeloid differentiation. Such cytokine-induced viral permissiveness coincides with cytokine-mediated CD34 removal from the cell surface [42].

We found that several SHREKs (PSGL-1, CD164, TIM-1, MUC1, and MUC4) inhibited Ha-CoV-2 infection. MUC1 and MUC4 are closely related mucins; MUC4 had the strongest inhibition among these SHREKs in our infection assay. Purified cell-free human breast milk mucins have been shown to possess anti-HIV activity [43]. A recent study has suggested that MUC4 expression plays a protective role in female mice in SARS-CoV infection [44]. In addition, in chikungunya virus (CHIKV) infection, the loss of MUC4 also results in augmented disease during early time points, indicating a possible broad role for MUC4 in viral infection and pathogenesis [44].

We found no inhibition of Ha-CoV-2 by CD43, CD34, PODXL1, PODXL2, and TMEM123 (Appendix A), although these proteins are potent inhibitors of HIV-1 (Figure 2B and Figure 3). It is possible that the difference may result from the different sites of viral assembly and budding. SARS-CoV-2 budding occurs mainly at the membranes of ER-Golgi intermediate compartment [45], whereas HIV-1 buds from the plasma membrane [46]. Because of the differences, it is possible that different sets of cellular proteins may be incorporated into HIV-1 and SARS-CoV-2.

Our identification of the SHREK family of proteins offers potential new antiviral therapeutics that may be developed through the induction and modulation of SHREK activities and the inhibition of viral antagonisms.

## 5. Patents

A provisional patent application pertaining to the results presented in this paper has been filed by George Mason University.

## Figures and Tables

**Figure 1 viruses-13-00832-f001:**
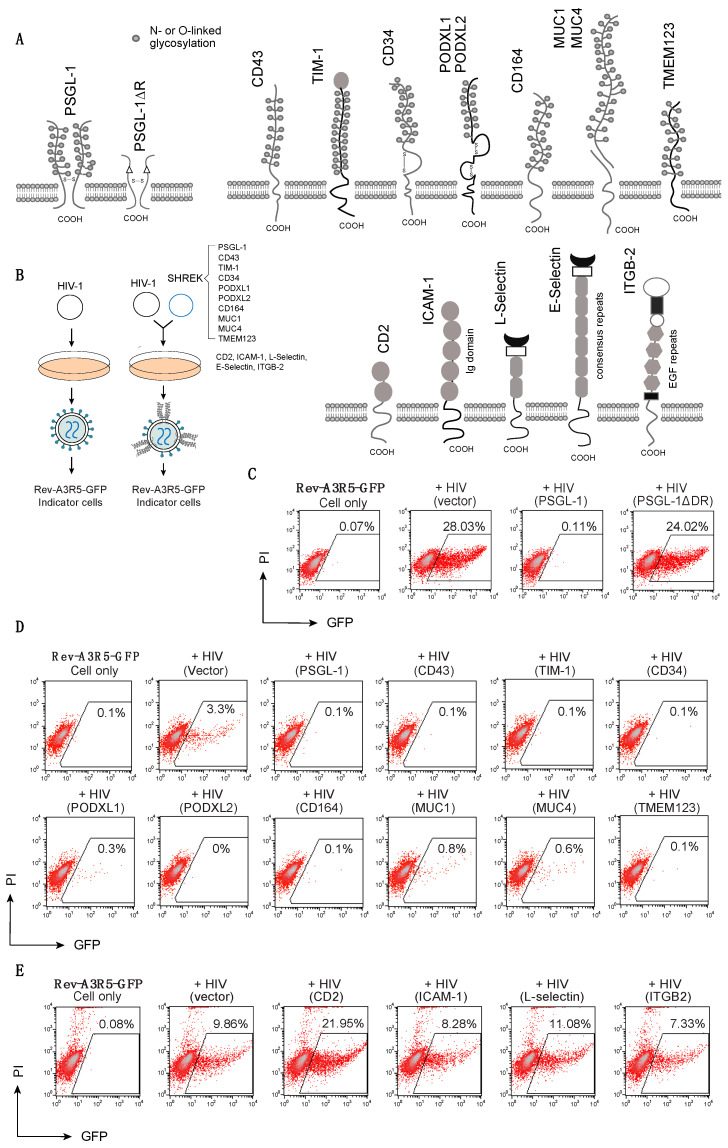
(**A**) Schematic of PSGL-1 and PSGL-1∆DR mutant, SHREK proteins, and other surface receptors used in this study. (**B**) Schematic of virion production in the presence of SHREK proteins in virus producer cells. (**C**) Requirement of the DR domain of PSGL-1 for blocking HIV-1 infectivity. HEK293T cells were cotransfected with HIV(NL4-3) DNA (1 μg) plus the vector expressing PSGL-1 or PSGL-1∆DR (400 ng). Virions were harvested at 48 h post-transfection and normalized for p24. Viral infectivity was quantified by infecting Rev-A3R5-GFP indicator cells with 300 ng p24 input of the viruses. HIV-1 replication was quantified by GFP expression at 72 h post infection. This experiment was repeated 3 times. (**D**) SHREK proteins inactivate HIV-1 virion infectivity. HEK293T cells were cotransfected with HIV(NL4-3) DNA (1 μg) plus each individual SHREK-expressing vector (100 ng for TIM-1, 500 ng for other SHREKs). Virions were harvested at 48 h and normalized for p24, and viral infectivity was quantified by infecting Rev-A3R5-GFP indicator cells with 35 ng p24 input of each virus. Shown are the percentages of GFP+ cells at 72 h post infection. The experiment was repeated 3 times. (**E**) For comparison, cells were also similarly cotransfected with HIV-1(NL4-3) DNA plus an empty vector or the vector expressing CD2, ICAM-1, L-selectin, or ITGB2. Virions were harvested and quantified by infecting Rev-A3R5-GFP indicator cells with 50 ng p24 input of each virus. The experiment was repeated 3 times.

**Figure 2 viruses-13-00832-f002:**
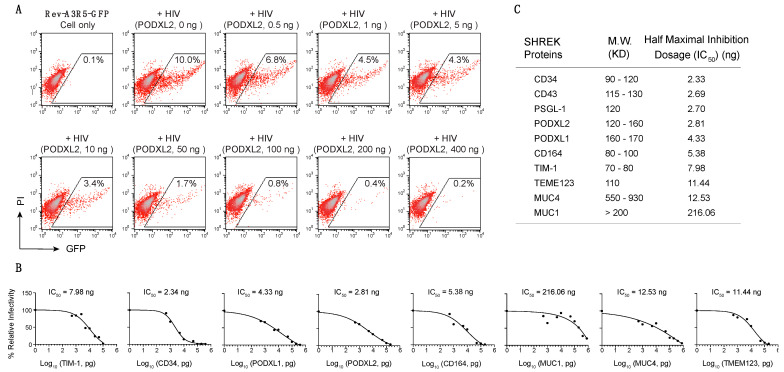
(**A**) Dose-dependent inhibition of HIV-1 infectivity by PODXL2. HEK293T cells were cotransfected with pHIV(NL4-3) DNA (1 μg) plus the PODXL2 expression vector (0.5 to 400 ng of DNA). Virions were harvested at 48 h and normalized for p24, and viral infectivity was quantified by infecting Rev-A3R5-GFP indicator cells. (**B**) Virions were assembled in the presence of each individual SHERK protein, and quantified for virus infectivity as in (**A**). The dose-dependent inhibition of HIV-1 curve by each individual SHREK protein was plotted using results from 2 independent experiments. (**C**) The 50% inhibition dosage (IC_50_) for each SHREK protein was calculated based on the averages from the two independent experiments.

**Figure 3 viruses-13-00832-f003:**
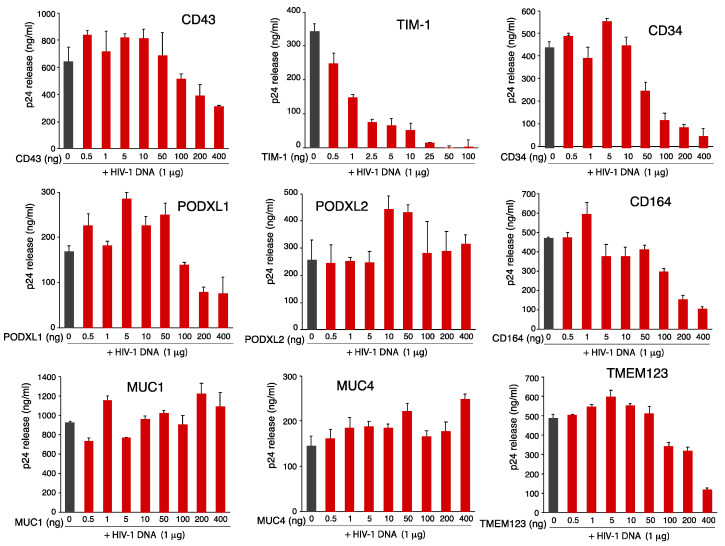
Effects of SHREK proteins on HIV-1 viral release. HEK293T cells were cotransfected with HIV(NL4-3) DNA (1 μg) plus each individual SHREK expression vector (0.5 to 400 ng of DNA). Virion release was quantified at 48 h post cotransfection by HIV-1 p24 ELISA. Data are represented as mean ± SD from ELISA triplicate. Experiments on the effects of TIM-1 were independently repeated 3 times (Appendix A), and experiments on effects of other SHREKs were independently repeated twice.

**Figure 4 viruses-13-00832-f004:**
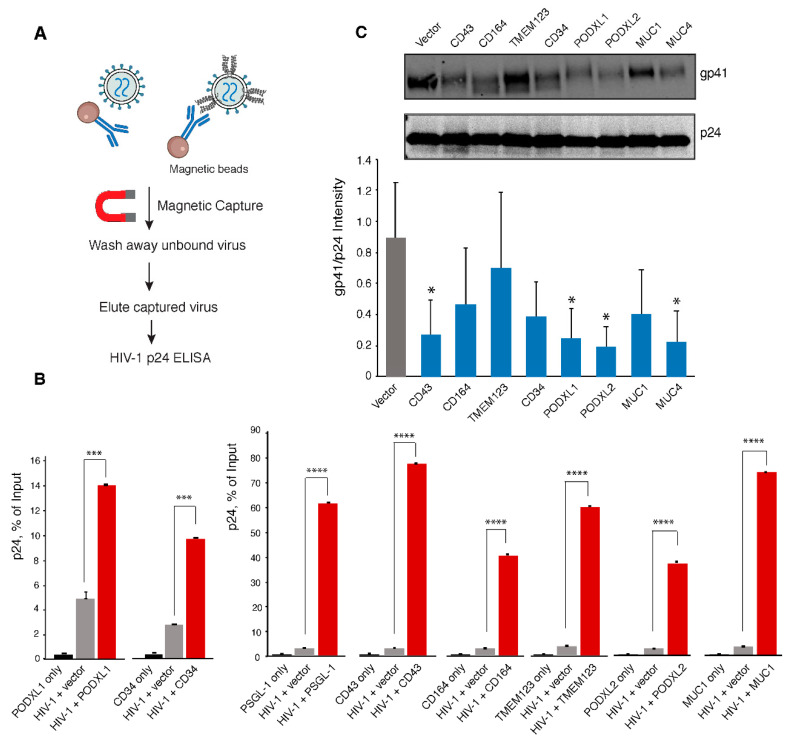
Virion incorporation of SHREK proteins and its effects on HIV-1 Env incorporation. (**A**) Schematic of the immune-magnetic capture assay used to detect SHREK proteins on HIV-1 particles. (**B**) HEK293T cells were cotransfected with pHIV-1(NL4-3) DNA (1 μg) plus each individual SHREK protein expression vector or empty vector (400 ng of DNA). As a control, cells were also transfected with only SHREK-expressing vector (400 ng). Empty vector was used to maintain an equal DNA concentration. For MUC1 and TMEM123, vectors expressing hemagglutinin-tagged TMEM123 or MUC1 were used. Supernatants were harvested at 48 h, normalized for p24, and incubated with magnetic beads coated with antibodies for each individual SHREK or hemagglutinin. Captured particles were washed, eluted, and quantified for the p24 levels with HIV-1 p24 ELISA. Data are presented as the percentage of input particles captured by the beads. The experiment was independently repeated 3 times, and the means ± SD from experiment triplicate are shown. (**C**) Effect of SHREK proteins on HIV-1 Env incorporation. HEK293T cells were cotransfected with HIV(NL4-3) DNA plus each individual SHREK protein expression vector or an empty vector. Particles were harvested at 48 h, and purified through a sucrose gradient. Virions were lysed and analyzed with Western blot using antibodies against HIV-1 gp41 and p24. Representative blots from 3 independent experiment repeats are shown. The band intensities were quantified from the three blots and normalized for p24. *p*-values were calculated using the two-tailed T-test. Significance values are indicated using asterisks as follows; * = *p* < 0.05, *** = *p* < 0.001, **** = *p* < 0.0001.

**Figure 5 viruses-13-00832-f005:**
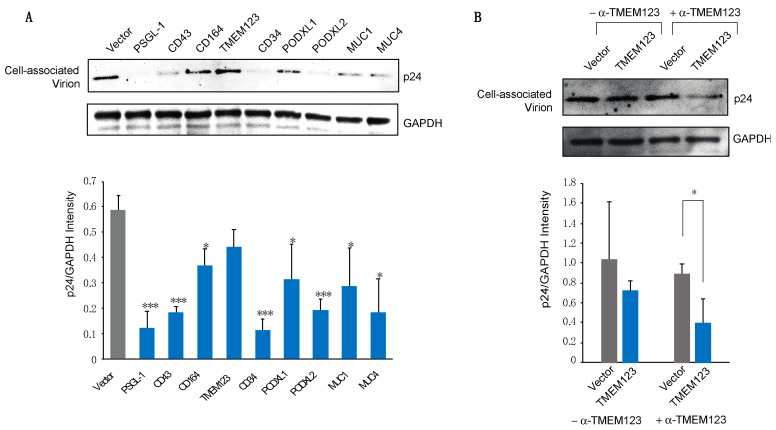
SHREK proteins block HIV-1 virion attachment to target cells. (**A**) Viral particles were produced by cotransfection of HEK293T cells with pHIV-1(NL4-3) DNA (1 μg) and each individual SHREK protein expression vector or an empty vector (500 ng). HIV-1 p24-normalized viral particles were then assayed for attachment to target HeLaJC.53 cells by Western blot for cell-bound p24. (**B**) Virions produced in the presence of the TMEM123 expression vector or the empty vector were assayed for attachment in the presence or absence of an anti-TMEM123 antibody. Representative blots from 3 experiment repeats are shown. The band intensities were quantified (for **A**,**B**) from the three blots and normalized with GAPDH. *p*-values were calculated using the two-tailed T-test. Significance values are indicated using asterisks as follows; * = *p* < 0.05, *** = *p* < 0.001.

**Figure 6 viruses-13-00832-f006:**
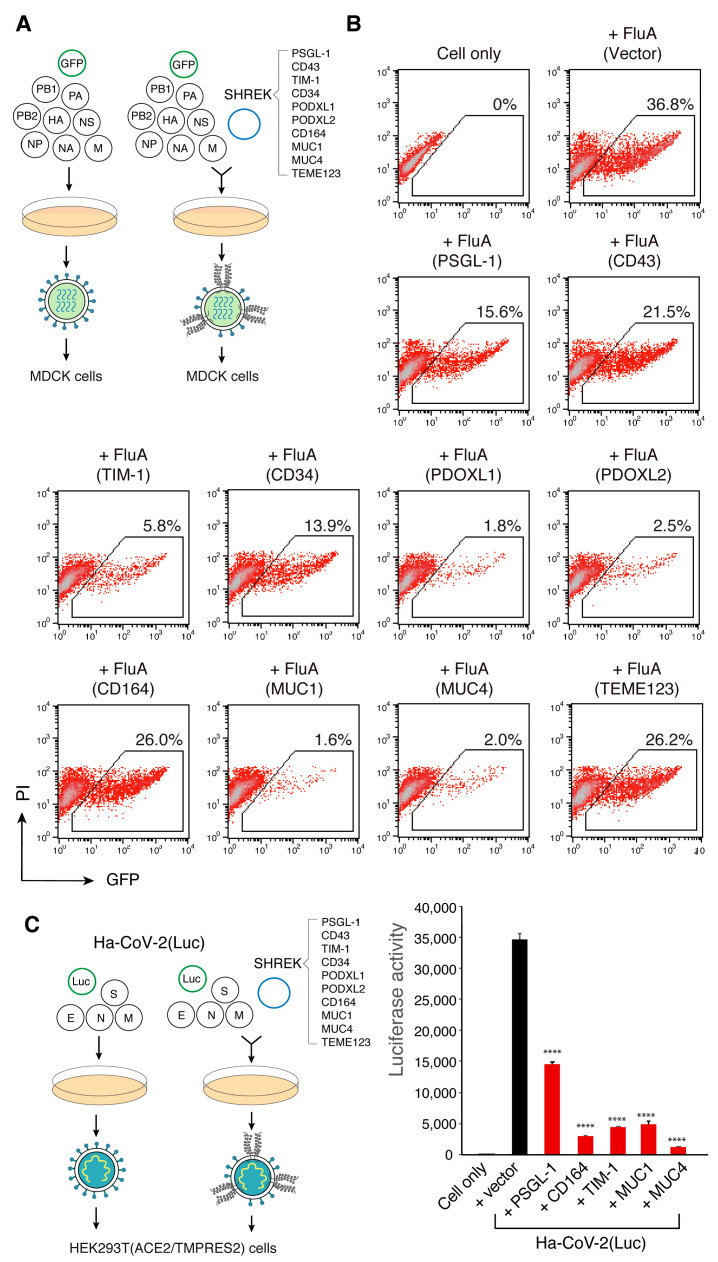
SHREK proteins are broad-spectrum host antiviral factors. (**A**) SHREK proteins inactivate influenza A virion infectivity. Influenza A-GFP reporter viruses were assembled by cotransfection of HEK293T cells with eight vectors expressing each of the segments of influenza A/WSN/33 (H1N1) (0.25 μg each), pHW-NA-GFP (ΔAT6) (1.5 μg), and each individual SHREK-expressing vector or an empty vector (0.5 μg). Viral particles were harvested at 48 h and used to infect target MDCK cells. This experiment was repeated 3 times. (**B**) Representative results from (**A**), showing GFP+ cells being quantified by flow cytometry following infection for 24 h. (**C**) SHREK proteins inhibit Ha-CoV-2 infection. Ha-CoV-2(Luc) particles were assembled in HEK293T cells by cotransfection of SARS-CoV-2 S-, M-, N- and E- expressing vectors (0.3 μg each), Ha-CoV-2(Luc) vector (1.2 µg), and each individual SHREK-expressing vector or an empty vector (1.6 μg). Virions were harvested, and used to infect HEK293T(ACE2/TMPRSS2) target cells. Luciferase activity was quantified at 18 h post infection. The experiment was repeated four times. Data are presented as means ± SD from experiment triplicate. *p*-values were calculated using the two-tailed T-test. Significance values are indicated using asterisks as follows; **** = *p* < 0.0001.

## Data Availability

The published article includes all data generated or analyzed during this study.

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
