# Peer review of "Identification of the SHREK Family of Proteins as Broad-Spectrum Host Antiviral Factors"

_viruses, 2021, doi:10.3390/v13050832_

Round 1
Reviewer 1 Report
Dabbagh and colleagues announce the identification of a family of proteins, named SHREK, that act as broad-spectrum host anti-viral factors. The SHREK family includes various human proteins sharing a common structure with PSGL-1, previously identified by other groups as capable to block HIV-1 infectivity. PSGL-1 acts via its incorporation in viral particles. Once incorporated, PSGL-1 is capable to interfere with HIV-1 attachment to target cells via a steric hindrance-based mechanism. They suggest that SHREK proteins act via similar mechanisms. Moreover, they propose that they are broad-spectrum antiviral proteins because they act also on the Influenza-A virus and SarsCoV2 pseudo-particles.
The manuscript is clearly presented. The methods are clear and the results are well described. It is potentially of interest, but some major controls are still missing, and in my opinion, they need to be presented to strengthen the conclusions.
The major issue I see with this work is that the authors only used 293T cells artificially expressing (or over-expressing) each of the SHREK proteins without testing the physiological level of expression in the cells. Moreover, none is shown about the stimuli that physiologically could increase SHREK proteins expression. Being anti-viral their expression is expected to increase following interferon stimulation, for instance. Is this true? For all of them?
It would be important to compare side-by-side cells naturally expressing the SHREK proteins and the 293T used to produce viral particles.
Further, in figure S2 it is shown that only 6-30% of 293T express the SHREK proteins. How many of them also co-express proviral DNA? Which dose of plasmid has been used to transfect 293T cells in this case? I am wondering how many cells produce viral particles without being also co-transfected with the SHREK-coding plasmid because this would impact the following tests.
Finally, it would be important to show that cells physiologically expressing the SHREK proteins, after being infected with HIV-1, produce viral particles that have incorporated SHREK proteins. This would exclude artificial incorporation due to the use of 293T cells.
Other points:
In most of the figures, it is not indicated how many times independent experiments have been performed. This should be corrected.
Various figures (e.g., figure 1C-E) only show representative results. Because experiments are expected to be performed at least 3 independent times, graphs showing the mean+-SD and statistical analysis should be presented for each figure.
For instance, in figure 1 the percentage of GFP+ cells in the control varies from 3.3% (panel D) to 28.03% (panel A). Thus, it is impossible to know if, for instance, ITGB2 is inactive or not.
Moreover, it is surprising to see that infectivity of viral particles is not increased in the presence of ICAM-1: this is in contrast with a large body of literature showing the positive effect of this protein on HIV-1 infectivity.
Figure 2 is meant to show the dose-dependent reduction of HIV-1 infectivity by SHREK proteins. Authors calculate the IC50 based on the amount (ng) of DNA used to transfect 293T cells. However, we do not know how many proteins are expressed in the cells after transfection (only one, unknown, dose of expression is shown in S2) and if all of them are equally expressed. Authors should find a manner to normalize protein expression (such as normalizing for actin) and re-calculate the values accordingly.
Figure 3: viral release should be calculated using the formula (Agp24 in the supernatant)/( Agp24 in the supernatant+Agp24 associated with transfected cells) to take into account eventual differences in transfection efficiency in the various conditions.
Figure 4: It would be important to indicate the percentage of viral particles of the initial input bound to the beads in this assay.
Figure 5: A quantification and normalization of the bands should be performed on the various independent experiments and showed as a graph.
Author Response
Reviewer 1:
We would like to thank the reviewer for the great efforts and the many constructive comments to help improving our manuscript. We have carefully considered these suggestions, and performed additional experiments to address these comments.
1) The major issue I see with this work is that the authors only used 293T cells artificially expressing (or over-expressing) each of the SHREK proteins without testing the physiological level of expression in the cells.
Response: to ensure that the virion inactivation phenotype does not purely result from over-expression of SHREK proteins in HEK293T cells, we used a range of vector dosages (0.5 – 500 ng) to co-transfect HEK293T cells. Most of the SHREK proteins such as PSGL-1, CD43, and CD34 can block HIV infectivity at very low vector dosages such as 1-5 ng. In contrast, a randomly selected protein (e.g. L-selectin) did not inhibit HIV infectivity even at 500 ng. To address the reviewer’s concern, we performed additional surface staining of HEK293T cells transfected with different dosages of CD43 and PSGL-1 (5, 50, and 500 ng), and compared the surface intensity of CD43 and PSGL-1 with that of two human T cell lines, Jurkat and CEM-SS. Both T cells express physiological levels of CD43 and PSGL-1. For CD43, at all dosages (5 - 500 ng), we did not observe higher expression of CD43 on transfected HEK293T cells. Even at 500 ng, the surface density of CD34 is comparable with that on T cells. For PSGL-1, we did not observe higher expression of PSGL-1 at low dosages (5 and 50 ng). Only at the high dose (500 ng), PSGL-1 expression in transfected HEK293T cells was higher. The new results are now presented in Supplemental Fig. S3. In addition, we also electroporated PSGL-1 into CEM-SS T cells. Over-expression of PSGL-1 in CEM-SS also led to reduction of virion infectivity, a phenotype similar to what we observed in HEK293T cells. The new results are presented in Supplemental Fig. S4.
2) Moreover, none is shown about the stimuli that physiologically could increase SHREK proteins expression. Being anti-viral their expression is expected to increase following interferon stimulation, for instance. Is this true? For all of them?
Response: We avoid naming SHREK proteins restriction factors, rather, we propose that SHREK proteins are a part of the innate immunity, such as the epithelium. Certain SHREK proteins such as PSGL-1 may also be a restriction factor and interferon induced, while others may simply be constitutively expressed on cell surface without the need for induction by interferon (e.g. CD34 on stem cells). Nevertheless, it has been reported that interferon can induce the expression of many of the SHREK proteins. Below is a partial list of previous publications. To address the reviewer’s suggestion, we added a new discussion with new references added (new third paragraph in Discussion):
TMEM123 induced by interferon:
- Rosengren, A. T., Nyman, T. A., Syyrakki, S., Matikainen, S., & Lahesmaa, R. (2005). Proteomic and transcriptomic characterization of interferon-alpha-induced human primary T helper cells. Proteomics, 5(2), 371–379. https://doi.org/10.1002/pmic.200400967
MUC4 induced by interferon:
- Andrianifahanana, M., Singh, A. P., Nemos, C., Ponnusamy, M. P., Moniaux, N., Mehta, P. P., Varshney, G. C., & Batra, S. K. (2007). IFN-gamma-induced expression of MUC4 in pancreatic cancer cells is mediated by STAT-1 upregulation: a novel mechanism for IFN-gamma response. Oncogene, 26(51), 7251–7261. https://doi.org/10.1038/sj.onc.1210532
MUC1 induced by interferon:
- Reddy, P. K., Gold, D. V., Cardillo, T. M., Goldenberg, D. M., Li, H., & Burton, J. D. (2003). Interferon-gamma upregulates MUC1 expression in haematopoietic and epithelial cancer cell lines, an effect associated with MUC1 mRNA induction. European journal of cancer (Oxford, England : 1990), 39(3), 397–404. https://doi.org/10.1016/s0959-8049(02)00700-1
CD164 upregulated by interferon:
- Khunger, A., Piazza, E., Warren, S., Smith, T. H., Ren, X., White, A., Elliott, N., Cesano, A., Beechem, J. M., Kirkwood, J. M., & Tarhini, A. A. (2021). CTLA-4 blockade and interferon-α induce proinflammatory transcriptional changes in the tumor immune landscape that correlate with pathologic response in melanoma. PloS one, 16(1), e0245287. https://doi.org/10.1371/journal.pone.0245287
- McLaren, P. J., Gawanbacht, A., Pyndiah, N., Krapp, C., Hotter, D., Kluge, S. F., Götz, N., Heilmann, J., Mack, K., Sauter, D., Thompson, D., Perreaud, J., Rausell, A., Munoz, M., Ciuffi, A., Kirchhoff, F., & Telenti, A. (2015). Identification of potential HIV restriction factors by combining evolutionary genomic signatures with functional analyses. Retrovirology, 12, 41. https://doi.org/10.1186/s12977-015-0165-5
CD43 induced by interferon:
- Zhou, H. F., Yan, H., Cannon, J. L., Springer, L. E., Green, J. M., & Pham, C. T. (2013). CD43-mediated IFN-γ production by CD8+ T cells promotes abdominal aortic aneurysm in mice. Journal of immunology, 190(10), 5078–5085. https://doi.org/10.4049/jimmunol.1203228
3) It would be important to compare side-by-side cells naturally expressing the SHREK proteins and the 293T used to produce viral particles.
Response: As we responded in point (1) above, we used a range of dosages (0.5 – 500 ng) of SHREK protein vectors. We performed additional experiments to compare the surface expression of CD43 and PSGL-1 on transfected HEK293T cells (doses 5, 50, and 500 ng) with that on jurkat and CEM-SS cells, which naturally express PSGL-1 and CD43. The new results are shown in Supplemental Fig. S3.
4) Further, in figure S2 it is shown that only 6-30% of 293T express the SHREK proteins. How many of them also co-express proviral DNA? Which dose of plasmid has been used to transfect 293T cells in this case? I am wondering how many cells produce viral particles without being also co-transfected with the SHREK-coding plasmid because this would impact the following tests.
Response: For figure S2, we used 400 ng of SHREK proteins to ensure that the surface staining signals are detectable. We co-transfected cells with 1000 ng of proviral NL4-3 DNA. We believed that at high-dosage of SHREK vectors (e.g. 400 ng), cells containing proviral DNA should also co-express SHREK DNA, and the inactivation of HIV virions is near 100%, whereas at low-dosages of SHREK vectors, some cells may produce virions in the absence of SHREK proteins, and the inactivation is only partial and SHREK dosage-dependent. We did not perform detailed quantification of how many cells produce virions in the absence of SHREK. Rather, we directly quantified the partial inactivation of virions released.
5) Finally, it would be important to show that cells physiologically expressing the SHREK proteins, after being infected with HIV-1, produce viral particles that have incorporated SHREK proteins. This would exclude artificial incorporation due to the use of 293T cells.
Response: This would be an experiment that we would like to perform for many of the SHREK proteins. Nevertheless, given that HIV infection of natural target cells normally leads to down-regulation of SHREK proteins (e.g. down-regulation of PSGL-1 by both Vpu and Nef). It would be technically challenging to detect the low-levels of SHREK proteins on infectious virions; it would need a major effort to produce highly purified particles to differentiate the low-level signals on virions from the low-level noise from contaminating extracellular vesicles.
Other points:
6) In most of the figures, it is not indicated how many times independent experiments have been performed. This should be corrected. Various figures (e.g., figure 1C-E) only show representative results. Because experiments are expected to be performed at least 3 independent times, graphs showing the mean+-SD and statistical analysis should be presented for each figure. For instance, in figure 1 the percentage of GFP+ cells in the control varies from 3.3% (panel D) to 28.03% (panel A). Thus, it is impossible to know if, for instance, ITGB2 is inactive or not.
Response: we have now revised our Figure legends to indicate how many times independent experiments have been performed. For the repeated infectivity experiments as shown in Figures such as those in Fig 1C-E, the differences were derived from different virus batches or different amounts of viral inoculum being used (35 ng to 300 ng of p24), or sometimes, different time-points being used (48 hours versus 72 hours) for quantifying GFP+ expression. Nevertheless, these independent experiments were highly reproducible, and conclusions were drawn independent of viral inoculum and time of analyses, although the percentages of GFP+ cells varied from experiments to experiments. Given the variability, results from these independent experiments are not applicable for averaging. However, in all independent experiments, an empty vector control was always used to quantify GFP+ expression in the absence of SHREK proteins. The inhibition or enhancement was always compared with this vector control. In the case of Fig. 1E, the independent control from the empty vector was 9.86%, whereas in the same experiment, the ITGB2 vector was 7.33%. The inhibition was minimal (26% inhibition) even with 500 ng of ITBG2. In Fig. 1D, the independent vector control from the empty vector was 3.3%, whereas the PSGL-1 vector was 0.1%. The inhibition was 99.7%.
7) Moreover, it is surprising to see that infectivity of viral particles is not increased in the presence of ICAM-1: this is in contrast with a large body of literature showing the positive effect of this protein on HIV-1 infectivity.
Response: we recognize that it has been well established that the enhanced infectivity of HIV-1 particles bearing ICAM-1 is mediated through interaction between ICAM-1 on HIV-1 particle and LFA-1 on target cells. In our system, we only observed some minor enhancement at certain low dosages of ICAM-1 (0.5 - 5 ng) (new Supplemental Fig. S1A). It is possible that the target cell, Rev-A3R5-GFP, used in our infectivity system, does not express high levels of active LFA-1, or there are some other adhesion molecules over-expressed on Rev-A3R5-GFP that may overshadow the role of ICAM-1/LFA-1. The infectivity system in the previous study (Fortin et al., 1997) used a different cell line for assays (1G5 T-cell line). To address the reviewer’s concern, we have performed additional dosage-dependent effects of ICAM-1 on HIV infectivity (new Supplemental Fig. S1A), and added new references to acknowledge previous studies. We also added a short discussion in Fig.S1 legend.
8) Figure 2 is meant to show the dose-dependent reduction of HIV-1 infectivity by SHREK proteins. Authors calculate the IC50 based on the amount (ng) of DNA used to transfect 293T cells. However, we do not know how many proteins are expressed in the cells after transfection (only one, unknown, dose of expression is shown in S2) and if all of them are equally expressed. Authors should find a manner to normalize protein expression (such as normalizing for actin) and re-calculate the values accordingly.
Response: We thank the reviewer for the suggestion. The comparison among SHREK proteins is only relative, and the IC50 values were based on the vector DNA not the protein levels, and thus we cannot conclude that one SHREK is more potent than the other. Although we had considered and attempted to normalize by protein levels, we realized that each commercial antibody reacts to target protein differently, and may need a common tag and anti-tag antibody to compare SHREK proteins. In addition, SHREK proteins may express at different subcellular locations, and presumably, only the surface-expressed SHREK proteins should be considered. Given the technique difficulty to accurately quantify surface receptors, we were not able to use protein levels for calculating IC50. To address the reviewer’s concern, we revised the text to reflect that the IC50 is only relative, and should not be used to compare the relative potency of individual SHREK proteins.
9) Figure 3: viral release should be calculated using the formula (Agp24 in the supernatant)/( Agp24 in the supernatant+Agp24 associated with transfected cells) to take into account eventual differences in transfection efficiency in the various conditions.
Response: We thank the reviewer for the suggestion. Given that almost all of these SHREK proteins did not inhibit HIV release (supernatant p24) at dosages below 100 ng, we assumed that at these dosages, these SHREK proteins unlikely inhibited p24 synthesis in cells. Thus, we did not perform extensive western-blot analysis to calculate the ratio. We expected that the normalized ratio would be the same as the results shown in Fig. 3.
10) Figure 4: It would be important to indicate the percentage of viral particles of the initial input bound to the beads in this assay.
Response: Figure 4 is updated to use % of input.
11) Figure 5: A quantification and normalization of the bands should be performed on the various independent experiments and showed as a graph.
Response: we have now included the quantification and normalization of blots from three repeats. Figure 5 and the legend have now been revised.
Reviewer 2 Report
- Need to add statistic significance in figures 2, 4, 6 or any figures that apply.
- IC50, 50 should be subscript, need to check it all through the whole manuscript.
- The authors proved that multiple SHREK proteins have antiviral activity through transfection of the plasmid encoding the proteins (overexpression), a knockout/knock down of these proteins would expected to see increased virus replication? It is worthy to investigate.
Author Response
Reviewer 2:
- Need to add statistic significance in figures 2, 4, 6 or any figures that apply.
Response: we have now added information on experiments repeats and p-values and statistic analysis to figures that apply.
- IC50, 50 should be subscript, need to check it all through the whole manuscript.
Response: we have now made the corrections.
- The authors proved that multiple SHREK proteins have antiviral activity through transfection of the plasmid encoding the proteins (overexpression), a knockout/knock down of these proteins would expected to see increased virus replication? It is worthy to investigate.
Response: we have performed shRNA knockdown of PSGL-1 in our previous publication (Fu et al, 2020, PNAS, 117:9537). We found that a slight reduction of PSGL-1 in human T cells led to an enhancement of HIV infectivity. We have attempted to knock down other SHREK proteins such as CD34. However, we found that most of the CD34+ stem cells have multiple SHREK proteins, and currently, it is technically challenging to knock down all these SHREK proteins to generate meaningful results.
Reviewer 3 Report
Dabbagh, et al. presents an interesting analysis of involvement of a large panel of SHREK family proteins as possible antiviral factors against enveloped viruses. The hypothesis is good and the data are interesting. However, there are many concerns regarding the presentation of the data that make this paper unable to be published in its present form. Some of the major issues seen by this reviewers are below:
In Figure 1, 2A and 6B, there is no indication that this experiment was completed in any number of replicates. Please identify the number of biological replicates. Ideally you would also include the averages of the replicates and the statistical analysis.
In Figure 2, if the dose response curve is plotted using data from 3 independent experiments, then the graphs should reflect that with proper error bars.
For Figure 3 and 4A/B, there is a lot of data but it is hard to see the significance. In terms of the legend saying “triplicate assays” does that mean one experiment with triplicates or three independent experiments? How are the data compared statistically? What statistical tests are used and what differences/p values are available. This is critical to see any conclusions from these graphs.
For Figure 4C, it is appropriate to have averages of the quantitation of the three Western blots included to give meaning the singular western blot. While p24 is consistent, the gp41 levels are definitely variable. Would it be appropriate to compare the two?
In Figure 5, it is definitely important to provide quantitation of replicates of the western blots and have the levels of p24 directly normalized to the GAPDH levels. The blots alone are not sufficient and must be presented with the graphs of the quantitation and the appropriate statistical analysis.
Figure 6C is very problematic. The authors try to frame the results to show that incorporation of the various proteins allow for virion production and that while in the virion, these SHREK proteins block infection to a target cell. However, I do not feel that in this system they have shown that the SHREK have not interfered with virion production. Quantitation of virion levels with a S ELISA, similar to a p24 ELISA, is required. In addition, the luciferase data has the same issues as above… is this an internal triplicate or are the experiments repeated on different days? Meaning is this a set of technical or biologic replications of the experiments. Also, there is no way to quantitate the meaning of the luciferase compared to cell viability. The authors need to show that the same level of virus is going in and that the luciferase is normalized to some level of viability of the cells.
Minor Note: As per the materials and methods, it is curious that the “HEK293T(ACE2/TMPRSS2)” cells do not require FBS to grow. Is that based on manufacturer’s instructions? Please add the concentrations of puromycin and hygromycin B that the media was supplemented with for these cells.
Author Response
Reviewer 3:
We would like to thank the reviewer for the constructive comments to help improving our manuscript. We have carefully considered these suggestions, and performed additional experiments or analyses to address these comments.
- In Figure 1, 2A and 6B, there is no indication that this experiment was completed in any number of replicates. Please identify the number of biological replicates. Ideally you would also include the averages of the replicates and the statistical analysis. In Figure 2, if the dose response curve is plotted using data from 3 independent experiments, then the graphs should reflect that with proper error bars. For Figure 3 and 4A/B, there is a lot of data but it is hard to see the significance. In terms of the legend saying “triplicate assays” does that mean one experiment with triplicates or three independent experiments? How are the data compared statistically? What statistical tests are used and what differences/p values are available. This is critical to see any conclusions from these graphs.
Response: We have now added the information of experimental replicates to the figure legends. For Fig. 1 and 6B, we have performed independent experiments for at least 3 times, and some (Fig. 1C) has been repeated for more than 3 times. We have now added the number of replicates in the figure legend. For Fig. 2, the dose-response cure, for the high dosages, we have repeated the experiments for 3 times to be sure that the inhibition phenotype is real. However, for the low dosages, we have repeated the experiments two times, largely because there were multiple dosages tested for each SHRKE, and the results have be very consistent, and showed a clearly dosages-dependent inhibition. For Fig.3, the inhibitory effect of TIM-1 on viral release was confirmed by 3 independent experiment repeats. However, for the other SHREK proteins, because there was a lack of inhibitory phenotype, and no clear inhibition of viral release was observed at dosages below 100 ng. We only repeated the largely negative results twice. For Fig. 4A/B, the experiments were independently repeated 3 times, and now we added error bars and statistic analyses to the Figure legend.
- For Figure 4C, it is appropriate to have averages of the quantitation of the three Western blots included to give meaning the singular western blot. While p24 is consistent, the gp41 levels are definitely variable. Would it be appropriate to compare the two?
Response: We have performed the Western blot 3 times, and now included a normalized bar graph from the quantification of the 3 western blots.
- In Figure 5, it is definitely important to provide quantitation of replicates of the western blots and have the levels of p24 directly normalized to the GAPDH levels. The blots alone are not sufficient and must be presented with the graphs of the quantitation and the appropriate statistical analysis.
Response: We have performed the Western blot 3 times, and now included a normalized bar graph from the quantification of the 3 western blots. We also added statistical analysis of these results.
- Figure 6C is very problematic. The authors try to frame the results to show that incorporation of the various proteins allow for virion production and that while in the virion, these SHREK proteins block infection to a target cell. However, I do not feel that in this system they have shown that the SHREK have not interfered with virion production. Quantitation of virion levels with a S ELISA, similar to a p24 ELISA, is required. In addition, the luciferase data has the same issues as above… is this an internal triplicate or are the experiments repeated on different days? Meaning is this a set of technical or biologic replications of the experiments. Also, there is no way to quantitate the meaning of the luciferase compared to cell viability. The authors need to show that the same level of virus is going in and that the luciferase is normalized to some level of viability of the cells.
Response: As we stated in the Discussion section that SHREK proteins can block viral infection through at least 3 mechanisms: blocking viral release, inhibiting the incorporation of viral attachment proteins (gp120 or S), or inactivating virion infectivity. For Fig. 6C, because our paper mainly focused on HIV, we did not study details of how some of the SHREK proteins can inhibit Ha-CoV-2 infection. Nevertheless, to accommodate the reviewer’s suggestion, we extended our studies and performed additional experiments by performing an S protein ELISA. The new results are now added as Supplemental Fig. S14. The new results showed that some SHREK proteins such as MUC4 may inhibit Ha-CoV-2 through blocking viral release or S protein incorporation, while others such as TIM-1 likely act through blocking Ha-CoV-2 particle infectivity. The lucifierase activity in Fig. 6C represents 3 independent biology experiment repeats. We also added statistical analysis. The Ha-CoV-2 infection assay is a quick infection assay with no detectable cytotoxicity as we described previously. Thus, we did not perform additional cytotoxicity control.
- Minor Note: As per the materials and methods, it is curious that the “HEK293T(ACE2/TMPRSS2)” cells do not require FBS to grow. Is that based on manufacturer’s instructions? Please add the concentrations of puromycin and hygromycin B that the media was supplemented with for these cells.
Response: We apologize for the omission. HEK293T(ACE2/TMPRSS2) cells do require FBS to grow. We have corrected the error. We also added concentration of puromycin and hygromycin B.
Round 2
Reviewer 1 Report
The authors answered my previous concerns and the manuscript is now suitable for publication